# Combination of Meropenem and Zinc Oxide Nanoparticles; Antimicrobial Synergism, Exaggerated Antibiofilm Activity, and Efficient Therapeutic Strategy against Bacterial Keratitis

**DOI:** 10.3390/antibiotics11101374

**Published:** 2022-10-07

**Authors:** Mohamed El-Telbany, Alzhraa Ali Mohamed, Galal Yahya, Aliaa Abdelghafar, Mahmoud Saad Abdel-Halim, Sameh Saber, Mohamed A. Alfaleh, Asmaa H. Mohamed, Fatma Abdelrahman, Hoda A. Fathey, Gehad H. Ali, Mohamed Abdel-Haleem

**Affiliations:** 1Microbiology and Botany Department, Faculty of Science, Zagazig University, Zagazig 44519, Egypt; 2Department of Microbiology and Immunology, Faculty of Pharmacy, Zagazig University, Zagazig 44519, Egypt; 3Department of Pharmacology, Faculty of Pharmacy, Delta University for Science and Technology, Gamasa 11152, Egypt; 4Department of Pharmaceutics, Faculty of Pharmacy, King Abdulaziz University, Jeddah 21589, Saudi Arabia; 5Vaccines and Immunotherapy Unit, King Fahd Medical Research Center, King Abdulaziz University, Jeddah 21859, Saudi Arabia; 6Center for Microbiology and Phage Therapy, Zewail City of Science and Technology, Giza 12578, Egypt

**Keywords:** *Pseudomonas aeruginosa*, biofilms, antimicrobial resistance, ZnO-NPs, gene expression, meropenem

## Abstract

*Pseudomonas aeruginosa* is an opportunistic gram-negative human pathogen that causes a wide range of infections, including nosocomial infections. Aside from the intrinsic and acquired antimicrobial resistance against many classes of antibiotics, *P. aeruginosa* can produce an extracellular polymeric matrix called “biofilm” that protects bacteria from antibiotics and harmful factors. Biofilm enables *P. aeruginosa* to develop chronic infections. This study assessed the inhibitory action of ZnO-nanoparticles against biofilms formed by multidrug-resistant *P. aeruginosa* strains. A collection of 24 clinical strains of *P. aeruginosa* were tested for their antimicrobial resistance against different antibiotics using the disk diffusion method. The antibiofilm activity of ZnO-NPs was assessed using the microtiter plate biofilm assay. The application of ZnO-NPs dramatically modulated the resistance profile and biofilm activity of *P. aeruginosa*. The combination of ZnO-NPs and meropenem showed synergistic antipseudomonal activity with lower MICs. The scanning electron microscope (SEM) micrographs revealed complete inhibition of biofilms treated with the meropenem–ZnO-NPs combination. Reduced expression of biofilm regulating genes *lasR*, *pslA,* and *fliC* was detected, reflecting the enhanced antibiofilm effect of ZnO-NPs. In vivo application of this antimicrobial mixture completely cured *P. aeruginosa*-induced keratitis in rats. Our findings represent a dual enhancement of antibacterial and antibiofilm activity via the use of meropenem–ZnO-NPs combination against carbapenem-resistant *P. aeruginosa* infections.

## 1. Introduction

*P. aeruginosa* is a Machiavellian pathogenic bacterium that causes refractory multidrug-resistant (MDR) infections, particularly in immunocompromised and hospitalized patients [1]. The increased incidence of *P. aeruginosa* nosocomial infections, the frequent emergence of MDR strains, and adaptive antimicrobial resistance during chronic infections represent a potential threat to human health worldwide [2]. 

Various *P. aeruginosa* infections are ascribed in pathogenesis to their capability to form biofilms, which are tight bacterial communities that can encase mucosal surfaces or intrusive devices [2]. Biofilms formed by *P. aeruginosa* create more favorable conditions for bacterial insistence, since embedded bacteria are more difficult to eliminate than planktonic bacteria [1]. 

Bacterial cells in biofilms can endure up to 1000 times higher antibiotic concentrations than planktonic bacterial cells [3]. Moreover, antibiotics have been found to be less functioning against biofilm-emergent bacteria [4]. Going through the genetic circuits regulating biofilm formation, studies revealed that the quorum sensing-related gene (*lasR*) is more abundant in biofilm-forming *P. aeruginosa* [5]. Additionally, the (*fliC*) gene plays a vital role in the initial attachment of *P. aeruginosa* when establishing the biofilm. Moreover, the (*fliC*) gene regulates the timing and rate of attachment and detachment but does not influence the biofilm architecture [6]. Besides the role of the (*fliC*) gene in biofilm attachment, the (*plsA*) gene has a significant role in biofilm adhesion, formation, and aggregation [7]. The (*plsA*) gene also has a role in the tolerance of *P. aeruginosa* biofilms to antibiotics [8]. Accordingly, there is an ultimate need to develop novel antimicrobial agents to combat and efficiently manage resistant pathogens [9]. Thus, most recent studies have moved towards investigation of new antimicrobial agents [10].

Carbapenems such as imipenem and meropenem, a class of β-lactam antibiotics, are considered as some of the best therapeutic options for combating MDR bacteria due to their high resistance to hydrolysis by plasmid- or chromosomally mediated β-lactamases produced by several pathogenic bacteria, such as *P. aeruginosa*, *Staphylococcus aureus*, and *Klebsiella* species [11]. Meropenem was found to be more efficient than imipenem due to the presence of a methyl group in the 1-position of the carbapenem moiety. This makes it more stable in vivo against inactivation by human renaldehydropeptidase-1 (DHP-1) [12]. Unfortunately, the resistance to both imipenem and meropenem was frequently detected among carbapenemase-producing organisms [13]. 

Nanomaterials with enhanced antimicrobial activities represent promising alternatives to traditional antibiotics and bypass the common antibiotic resistance strategies owing to the fact that most antibiotic-resistance mechanisms do not match up with the pathways wherein nanomaterials act [14]. Among nanomaterials, the metal oxide nanoparticles, such as nanoparticles of zinc oxide (ZnO), have recently received greater consideration because of their antimicrobial activity against both Gram-positive and Gram-negative spores [15,16,17,18]. ZnO-NPs are also recognized as being nontoxic, bio-safe, and biocompatible [19]. ZnO-NPs have prominent antimicrobial activity through interference with the cell wall and cell membrane synthesis, resulting in increased membrane permeability and nanoparticle internalization which generates reactive oxygen species (ROS) and leads to elevated membrane lipid peroxidation, loss of proton motive force, and mitochondria dimness due to uptake of liquified Zn ions, intracellular discharge, the liberation of oxidative stress gene expression, subsequent inhibition of cell growth, and cell decease [20,21,22,23,24,25,26]. A recent study denoted that lowering the virulence efficacy of *P. aeruginosa* via the quorum-quenching action of ZnO-NPs is caused either by inhibiting quorum-sensing (QS) molecules production or through interactions with the QS molecules, making them unidentifiable to cell machinery [27].

Combinations of ZnO-NPs and antibiotics, such as aminoglycosides–cephalosporins and aminoglycosides–beta-lactams, have been reported as promising strategy against numerous MDR bacteria [28]. Considering the severity of *P. aeruginosa* biofilm-driven infections and the elevated antibiotic and antimicrobial resistance rates, herein we studied the antibiofilm inhibitory properties of ZnO-NPs against pre-formed *P. aeruginosa* biofilms. We also investigated the effects of the meropenem–ZnO-NPs combination against clinically isolated *P. aeruginosa* strains. In this presented study, we boost the antibacterial activity of meropenem antibiotics by coupling them with a low concentration of ZnO-NPs so they can be used as a bio-control agent against *P. aeruginosa* bacteria and biofilms. 

## 2. Results

### 2.1. Antibiotic Susceptibility

*P. aeruginosa* strains were tested for their susceptibilities to 10 antibiotics routinely prescribed in human medicine using the disk diffusion method. A remarkable variation in the frequencies of antibiotic resistance was detected among isolates. The obtained results (Table 1) were compared to the documented standard diameter of inhibition zones for each antibiotic in CLSI-2021 [29]. All strains (100%) were meropenem-resistant and 96% of tested strains were resistant to levofloxacin, ceftazidime, gentamycin, and piperacillin. About 91% of tested strains were resistant to amikacin and imipenem, while 83% were resistant to aztreonam, 79% exhibited resistance to cefepime, and 75% were resistant to norfloxacin. Overall, most of the tested strains displayed resistance to the two tested carbapenem antibiotics (imipenem and meropenem).

### 2.2. Synergy between ZnO-NPs and Meropenem

The MICs of ZnO-NPs and meropenem decreased considerably when used in combination compared to their independent MICs (Appendix A). The MIC of ZnO-NPs was 64 µg/mL, and this concentration falls to 8 µg/mL when used in combination with meropenem (Table 2). On the other hand, meropenem alone has a MIC of 512 µg/mL and, when combined with ZnO-NPs, this value dropped to 4 µg/mL (Table 2). Importantly, the synergistic effect of ZnO-NPs on the antipseudomonal activity of meropenem was assessed against *P. aeruginosa* PU15 through determination of the fractional inhibitory concentration (FIC) index (Table 2). MIC values for both meropenem and ZnO-NPs were reduced when combined, to provide FIC values less than 0.5, indicating noticeable synergy. 

### 2.3. Antibiofilm Activity of ZnO-NPs

Biofilm formation was investigated in vitro using the crystal violet staining technique. As represented in (Figure 1), the biofilms formed by tested strains were assessed by measuring the optical density (O.D) at 570 nm. The high readings refer to the thickness of the formed biofilm. All test strains were able to form a biofilm matrix. *P. aeruginosa* PU15 was the strain with the thickest biofilm (O.D 570 = 0.5). This isolate was used for subsequent experiments.

Next, we evaluated the antibiofilm activity of ZnO-NPs and a 70 to 85% decrease in the biomass of biofilms formed by most strains was detected, confirming the powerful antibiofilm activity of ZnO-NPs (Figure 1). Interestingly, the biofilm formed by *P. aeruginosa* PU15 lost 80% of its biomass after treatment with 50 µg/mL of ZnO-NPs. 

### 2.4. Macroscopical and Microscopical Examination of Biofilm Disruption 

Antibiofilm activity of ZnO-NPs was further assessed using the Congo red agar assay by monitoring the appearance of black colonies, which indicate the production of exopolysaccharides and the formation of biofilms. *P. aeruginosa* PU15 was streaked on GRA in the absence of ZnO-NPs and the bacteria formed black colonies (indicating biofilm formation), whereas bacteria cultured after treatment with ZnO-NPs formed red colonies and small black colonies. Red colonies show that biofilm formation by PU15 was impeded by ZnO-NPs, although bacterial viability was not highly affected (Figure 2a).

SEM was used to examine the morphology of biofilms from the PU15 strain and to monitor anatomical changes in the biofilm upon treatment with ZnO-NPs alone, as well as when combined with meropenem. Results showed that *P. aeruginosa* PU15 biofilms formed on glass coverslips for 24 h from clumped and aggregated cells. The addition of 8 μg/mL of ZnO-NPs was able to reduce the size of biofilm clusters and control bacterial growth. Interestingly, the combination of ZnO-NPs and meropenem at MIC (8 and 4 μg/mL, respectively) was able to dramatically eradicate the biofilm (Figure 2b).

### 2.5. Expression of Biofilm Genes

We then evaluated the expression profile of genes regulating biofilm formation, before and after treatment with ZnO-NPs, via RT-PCR. The results showed that expression of the tested genes significantly decreased in treated samples compared to untreated samples. A two- to four-fold reduction in *lasR*, *fliC*, and *pslA* gene expression was observed after treatment with ZnO-NPs, as shown in (Figure 3). 

### 2.6. Effect of ZnO-NPs and Meropenem on the Area % of Corneal Opacity as a Measure of Corneal Lesions

Pictures of infected corneas were processed using Imagej 1.52i software (NIH, Bethesda, MD, USA). As shown in Figure 4a, the Imagej-processed photographs designate the areas of focal lesions. As a measure of the clinical presentation of corneal *P. aeruginosa* infection, the Area % of corneal opacity was calculated. This method was performed to minimize bias and errors in assessments of clinical presentation scores by an ophthalmologist, even when blinded [30]. Therefore, more accurate determinations are assured. In this regard, treatment of keratitis with the meropenem–ZnO-NPs combination significantly attenuated infection in rats compared to treatment with meropenem alone. This is indicated by the significant reduction in the Area % of corneal opacity value in the rat group treated with the combination compared to the rat group solely treated with meropenem (Figure 4b).

## 3. Discussion

Due to the excessive and imprudent use of antibiotics in human and veterinary medicine, the post-antibiotic era, when all of our discovered antibiotics become outmaneuvered by MDR bacteria or superbugs, is underway [3]. Among the most serious bacterial infections are those caused by *P. aeruginosa*, which is a highly volatile genome capable of developing fast resistance and a wide range of recalcitrant infections, including bloodstream infections, pneumonia, and both community-acquired and hospital-acquired diseases [31,32]. The resistance-developing characteristics of *P. aeruginosa* dominate in mixed infection communities and cause various infections (from skin to bloodstream and lung infections), which is why it is denoted by the World Health Organization (WHO) as one of the most critical pathogens requiring new approaches to control [32]. The severity of *P. aeruginosa* infections comes from their ability to form biofilms that can cover mucosal surfaces [2], as well as their ability to make embedded bacterial cells more difficult to eliminate [1]. Antibiotic susceptibility tests on 24 *P. aeruginosa* strains revealed that all tested strains exhibit resistance to meropenem. In addition, 91% of the strains were imipenem-resistant. This resistance pattern against two of the most commonly used carbapenem antibiotics led us to investigate potential improvement in their activity by combining them with other non-antibiotic compounds that can disrupt bacterial biofilm.

Numerous studies have elucidated the possibility of strengthening and improving the activity of certain antibiotics by combining them with nanoparticles, either additively or synergistically [33]. The antimicrobial mechanism of action of NPs is generally assigned to one of three mechanisms: oxidative stress induction, [20] metal ion release [21], or non-oxidative mechanisms [22]. In agreement with other studies [28,34,35,36,37,38,39], combining meropenem with ZnO-NPs showed striking antimicrobial synergism against *P. aeruginosa*. It is known that carbapenems act as inhibitors of the peptidase domain of penicillin-binding proteins (PBPs), and carbapenems can impede peptide cross-linking as well as other peptidase reactions. Cell wall formation is a three-dimensional process where formation and autolysis occur at the same time when PBPs are inhibited; as autolysis continues, it eventually leads to weakness in peptidoglycan and cell rupture [40]. It is also known that ZnO-NPs can inhibit bacterial growth by permeating into the cell membrane; moreover, ZnO-NPs can generate stress, leading to damage to lipids, carbohydrates, proteins, and DNA which eventually disrupts vital cellular functions [41]. 

All the tested strains in our study efficiently formed dense biofilms, and *P. aeruginosa* PU 15 showed the highest biofilm intensity. For combating biofilm-forming bacteria, there are three universal strategies: (i) evading cell-surface adhesion, (ii) distracting biofilm development and/or affecting biofilm architecture in order to increase antimicrobial penetration, and (iii) distressing biofilm maturation and/or inducing its dispersal and degradation [23,24]. In line with other studies [42], the substantial reduction in biofilm density (up to 85%) was observed upon treatment with sub-MIC doses of ZnO-NPs. Biofilm disruption was checked microscopically, and the SEM examination showed that the meropenem–ZnO-NPs combination was able to completely inhibit the pre-formed biofilm of *P. aeruginosa* PU15.

Biofilm formation is regulated at the molecular level through a package of biofilm genes (*lasR*, *fliC*, and *pslA*) [43], a reduction in biofilm density is associated with diminished expression of biofilm genes [44,45], and the quantification results of the expression of biofilm genes (*lasR*, *fliC*, and *pslA*) showed a significant decrease in expression. Taken together, the mixture of meropenem and ZnO-NPs downregulates biofilm gene expression, which provides reasonable explanation for the reduced biofilm density. Additionally, the small size of the ZnO-NPs increases their ability to penetrate the biofilm matrix and leads to more enhanced antibiofilm activity [28], which explains the complete removal of biofilm in some fields during SEM examination. 

Finally, regarding benefits from the synergistic antimicrobial and antibiofilm effect of the meropenem–ZnO-NPs combination, the application of this mixture successfully protected the cornea rat model from pseudomonas-induced keratitis, which indicates the clinical significance of this combination, especially against bacterial keratitis.

## 4. Materials and Methods

### 4.1. Bacterial Strains

A total of 24 *P. aeruginosa* isolates, including standard strain *P. aeruginosa* PU15 (accession number: LC514698), were obtained from Mohamed et al. [46]. Twenty-three strains were genetically confirmed using the forward primer 5′-GGGGGATCTTCGGACCTCA-3′ and reverse primer 5′-TCCTTAGAGTGCCCACCCG-3′ (Midland Certified Reagent Company, Midland, TX, USA). The *P. aeruginosa* PU15 strain was used as a positive control. Stocks were maintained in 20% (*v*/*v*) glycerol at −80 °C until needed. Bacterial strains were grown on Cetrimide agar (Sigma-Aldrich, St. Louis, MO, USA) overnight at 37 °C [47]. 

### 4.2. Antimicrobial Susceptibility Testing 

Ten antibiotics (purchased from Oxoid Ltd., Basingstoke, UK), namely imipenem (10 µg), meropenem (10 µg), levofloxacin (5 µg), ceftazidime (30 µg), gentamycin (10 µg), amikacin (30 µg), aztreonam (30 µg), piperacillin (100 µg), cefepime (30 µg), and norfloxacin (10 µg), were selected for the antibiotic susceptibility test with *P. aeruginosa* using the disk diffusion susceptibility method [48]. Overnight bacterial culture in Tryptic Soy Broth (TSB) (Difco, San Diego, CA, USA) was adjusted to 0.5 McFarland turbidity standards. Subsequently, 0.1 mL of bacterial suspension was spread, using sterile swabs, on Mueller–Hinton agar plates (HI Media Lab. Pvt. Ltd. Ref. M173). Duplicate plates were prepared for each isolate. The antibiotic discs were placed on the agar plates (within 15 min of the inoculation) using sterile forceps, to apply the discs at a distance of 2 cm apart from each other, and incubated for 16–18 h at 37 °C. After incubation, inhibition zones were visible and measured with a ruler, with the measurements recorded in mm [49]. 

### 4.3. Biofilm Assay Method

A biofilm assay was conducted for the 24 *P. aeruginosa* strains to compare their biofilm potential using the microtiter plate biofilm assay [50]. A total of 200 µL of each strain, with a concentration of 10^8^ CFU/mL, was moved into a 96-well polystyrene flat-bottomed microtiter plate (Costar; Corning Inc., Corning, NY, USA) and incubated for 48 h at 37 °C. TSB media without any bacteria were used as a negative control. After incubation, the plate was washed three times with sterile saline to remove the planktonic cells and the wells were left to dry for 1 h at 37 °C. About 125 μL of 0.1% crystal violet solution was added to each well and incubated for 20 min at room temperature in order to quantify the biofilm biomass, before then being washed three times with water. A total of 200 μL of 95% ethanol was used to dissolve the stained biofilm, which was then applied to an ELISA plate reader to measure the optical density (O.D) at 570 nm.

### 4.4. Preparation of Antimicrobial Agents (Meropenem and ZnO-NPs)

(a) Meropenem stock solution (20 µg/mL) was prepared by dissolving 0.02 mg of meropenem trihydrate (Meronem^®^ IV 500 mg, Pfizer, New York, NY, USA) in 1 mL of sterile double-distilled water. This solution was freshly prepared before each experiment.

(b) The ZnO-NPs suspension was prepared immediately before each experiment. After weighing the ZnO nanopowder (677450- CAS NO:1314-13-2, Sigma, USA, ≤50 nm), 10 mg of the powder was added to a tube containing 1 mL of sterile distilled water, which was then subjected to sonication in a pan sonicator (Branson, MO, USA) for at least 1 h until the uniformly suspended suspension was obtained. The resulting suspension was subjected to a vigorous vortex before each use.

### 4.5. Detection of the Antibiofilm Activity of ZnO-NPs

The antibiofilm activity of ZnO-NPs was monitored using the microtiter plate biofilm assay [51], as described previously. A total of 100 µL of each strain, with a concentration of 10^8^ CFU/mL, was inoculated into a 96-well polystyrene flat-bottomed microtiter plate, followed by the addition of 200 μL per well of ZnO-NPs (50 μg/mL); wells left without any treatment were used as a positive control. Each treatment was applied in triplicate in three wells. The plate was incubated for 24 h at 37 °C without shaking. After incubation, the plates were washed three times to remove the planktonic cells, and wells were stained with 0.1% crystal violet solution for 20 min. The wells were then washed three times with water, and 200 μL of 30% acetic acid was applied to dissolve the stained biofilm. The change in absorbance was evaluated by measuring the optical density (O.D) at 570 nm in order to detect the biofilm inhibition effect of ZnO-NPs against *P. aeruginosa*.

### 4.6. Determination of MICs of Meropenem and ZnO-NPs (Microdilution Method)

The broth microdilution method was conducted according to the recommendations of the Clinical and Laboratory Standards Institute [29] to determine the minimum inhibitory concentrations (MICs) of meropenem and ZnO-NPs against *P. aeruginosa* PU15. Double serial dilutions of meropenem and ZnO-NPs were prepared using 96-well microplates (Costar; Corning Inc., Corning NY, USA) with cation-adjusted Mueller–Hinton Broth (CAMHB, HI Media Lab. Pvt. Ltd. Ref. M1657). The concentration ranges of ZnO-NPs and meropenem were 1–128 and 1–512 μg/mL, respectively. The bacterial suspension was adjusted according to the 0.5 McFarland standards and added to each well (final bacterial concentration: ~10^5^ CFU/mL). Additionally, the control of bacterial growth (CAMHB + bacteria) and medium sterility (CAMHB) were tested on each test plate. Microtiter plates were incubated at 35–37 °C for 18–24 h. The MIC values were determined by comparing the growth density in the wells containing antibiotics with those in the control wells used in each test set. Each test was performed in triplicate.

### 4.7. Evaluation of Synergistic Antibacterial Activity (Checkerboard Microdilution Assay)

The activity of the ZnO-NPs–meropenem combination against *P. aeruginosa* PU15 was assessed using the checkerboard synergy technique [39] based on microdilution. The competence of the combination of the two antimicrobials was tested using a 96-well microtiter plate and CAMHB as a medium. Serial 2-fold dilutions of ZnO-NPs and meropenem were mixed in each well of a 96-well microtiter plate so that each row (and column) contained a fixed amount of one agent and increasing amounts of the second agent. Stock solutions of ZnO-NPs and meropenem were diluted in an appropriate volume of CAMHB. Bacterial suspension stock was adjusted to ~10^5^ CFU/mL. The resulting plate presents a pattern in which every well contains a unique combination of concentrations between the two agents. The concentrations of meropenem ranged from 0.5 to 512 µg/mL, while ZnO-NPs concentrations ranged from 1 to 64 µg/mL. Bacterial control (CAMHB + bacteria) and medium sterility control (CAMHB) for each plate were measured. The plates were incubated at 35–37 °C for 18–24 h. Each experiment was conducted three times. The fractional inhibition concentration (FIC) of both antimicrobials was calculated to be able to infer the results according to the following formulas:
FICA = ((MIC of A in Combination)/(MIC of A alone))
FICB = ((MIC of B in Combination)/(MIC of B alone))
FIC index (FICI) = FICA + FICB

FICI ≤ 0.5 was taken as synergism; 0.5 ≤ FICI ≤ 4 was taken as indifference; and FICI > 4 was taken as antagonism [48].

### 4.8. Congo Red Agar Assay (CRA)

A Congo red agar assay (CRA) was performed, according to the method given by [52], to evaluate the role of ZnO-NPs on biofilm formation by *P. aeruginosa*. Brain heart infusion broth (Sigma-Aldrich, St. Louis, MO, USA), 20 g/L of agar powder, and 8% Congo red indicator were prepared and autoclaved at 121 °C for 15 min. *P. aeruginosa* PU15 treated with the MIC of ZnO-NPs was streaked. *P. aeruginosa* PU15 without any treatment was streaked and used as a control. The two groups (control and treatment) were incubated overnight at 37 °C. After incubation, both plate groups were compared for colony color. Black colonies with a dry crystalline uniformity indicate biofilm formation.

### 4.9. SEM

The antibiofilm activity of ZnO-NPs and the meropenem–ZnO-NPs combination against the biofilm formed by *P. aeruginosa* PU15 was examined using SEM, as described previously [53]. Briefly, a sterile coverslip was added to each well of a sterile 24-well microtiter plate (Sigma) using sterile forceps, with 200 μL of sterile brain heart infusion broth (BHIB, HI Media Lab. Pvt. Ltd. Ref. MV210) and 200 μL of a bacterial culture of *P. aeruginosa* PU15 (0.2 O.D at 600 nm) then added to each well. After incubation at 37 °C for 24 h, medium and planktonic bacteria on the glass coverslips were removed by washing the plate three times with PBS. Treatments with ZnO-NPs (8 μg/mL), the meropenem–ZnO-NPs combination (4 μg/mL and 8 μg/mL, respectively), and the control saline-treated sample were applied separately (each on a coverslip) and incubated at 37 °C for 24 h. Afterward, coverslips were washed gently three times with PBS and fixed by 2.5% glutaraldehyde for 2 h. Fixed slides were subsequently washed again with PBS, and dehydrated through serially graded ethanol solutions. Slides were dried and gold-coated before imaging and examined using a JEOL scanning microscope (JSM 6510 lv, Peabody, MA, USA). The SEM images were obtained at the Electronic Microscope Unit, College of Agriculture, Mansoura University, Mansoura, Egypt.

### 4.10. Quantitative Reverse Transcriptase Polymerase Chain Reaction RT-PCR

RT-PCR was used to assess the expression of biofilm genes (*lasR*, *fliC*, and *pslA*) in *P. aeruginosa* PU15 both treated and untreated with ZnO-NPs. RNA was extracted from samples using a QIAamp Rneasy Mini kit (Qiagen, Hilden, Germany, GmbH). A volume of 200 µL of each sample was added to 600 µL of RLT buffer containing 10 μL of β-mercaptoethanol per 1 mL and incubated at room temperature for 10 min. One volume of 70% ethanol was added to the lysate, and the steps were completed according to the Total RNA Purification protocol of the QIAamp Rneasy Mini kit (Qiagen, Hilden, Germany, GmbH). The primers used in this step (Table 3) were supplied from Metabion (Germany) and the amplification curves of the primers are shown in Appendix A. Primers were utilized in a 25 µL reaction containing 10 µL of the 2× HERA SYBR^®^ Green RT-qPCR Master Mix (Willowfort, UK), 1 µL of RT Enzyme Mix (20×), 0.5 µL of each primer at 20 pmol concentration, 5 µL of water, and 3 µL of RNA template [54]. The reaction was performed in a step one real-time PCR machine [55]. Amplification curves and ct values were determined by the step one software. To estimate the variation of gene expression in the RNA of the different samples, the CT of each sample was compared to that of the positive control group according to the “ΔΔCt” method stated by Yuan et al. [56] using the following ratio: (2^−ΔΔct^).

### 4.11. Animal Study

Adult male SD rats (220 ± 20 g) were obtained from Delta University for Science and Technology, Egypt. All animal care and experimental procedures were approved by the Institutional Animal Care and Use Committee (Approval Number: FPDU17220/2) and carried out in accordance with relevant guidelines and regulations.

#### 4.11.1. Induction of Keratitis

After anesthetization with an intraperitoneal ketamine (50 mg/kg)-xylazine (10 mg/kg) mixture [59,60], the right eye cornea of each rat was scarred using a 27-gauge needle (three 1-mm scratches). A volume of 10 µL of 0.6% acetylcysteine was pipetted onto the cornea to break up the tear film and washed with normal saline solution. This was followed by the application and even distribution of a 10 µL suspension containing (5 × 10^8^ CFU) of *P. aeruginosa* onto the wounded cornea. The left eyes of rats served as a blank eye and were scarred by the same pattern but not subjected to infection [61]. This protocol induces inflammatory keratitis, wherein inflammatory cells infiltrate different corneal layers in response to an infectious exogenous agent. The ensuing modulation of inflammation and immune signaling exacerbates cellular damage [62] that necessitates effective bacterial clearance. 

#### 4.11.2. Experimental Design

For rapid induction of keratitis, thirty rats were divided into 5 groups (6 per group) as follows: the control group, in which uninfected rats served as control animals; normal rats, which served as control animals; untreated rats that were infected with *P. aeruginosa*; rats that were infected with *P. aeruginosa* and treated with ZnO-NPs dispersion; rats that were infected with *P. aeruginosa* and treated with meropenem; and rats that were infected with *P. aeruginosa* and treated with meropenem + ZnO-NPs. Administration was started two days after infection. The infection was established and confirmed by an ophthalmologist who was blinded to the treatment protocol. A clear, visibly cloudy appearance of corneas was observed on the second day after *P. aeruginosa* infection. Treatment was continued for another 3 days.

### 4.12. Statistical Analysis 

Using GraphPad Prism software version 8.0.2, statistical analysis was conducted (GraphPad Software Inc., La Jolla, CA, USA). Two-tailed unpaired T-tests were employed to analyze significance in Figure 1 and Figure 3, and differences between groups in Figure 4 were analyzed by one-way analysis of variance followed by a post-hoc Tukey test. A value of *p* ≤ 0.05 was considered to indicate statistical significance.

## 5. Conclusions

In conclusion, our study revealed the potential application of combination therapies of NPs with antibiotics, as opposed to the use of monotherapies which may lead to more resistance and, consequently, treatment failures. Our results earmark the improved antibacterial and antibiofilm activity of the meropenem–ZnO-NPs combination, representing a successful strategy for overcoming the emergent resistance of *P. aeruginosa* against carbapenems. 

## Figures and Tables

**Figure 1 antibiotics-11-01374-f001:**
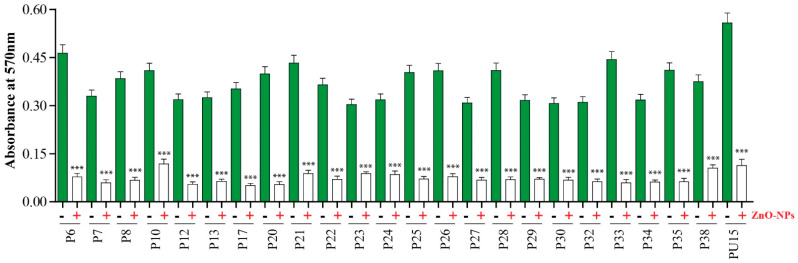
Changes in biofilm density upon treatment with ZnO-NPs. Colorimetric measurement of biofilm formation by *P. aeruginosa* isolates before (green columns) and after treatment with ZnO-NPs (white column). Results are expressed as Mean ± SEM from three independent experiments. *** *p* ≤ 0.001.

**Figure 2 antibiotics-11-01374-f002:**
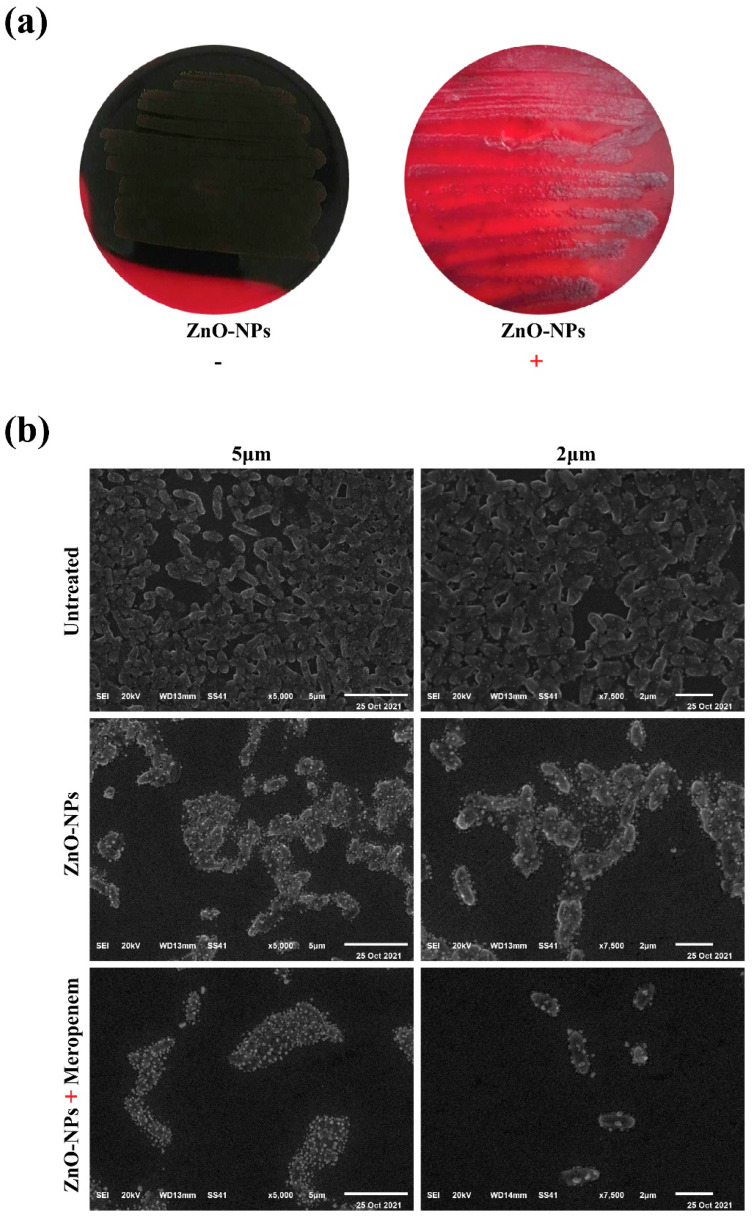
Phenotypic and SEM examination of antibiofilm activity. (**a**) The growth of *P. aeruginosa* PU15 (untreated or treated with ZnO-NPs) on Congo red agar. (**b**) SEM micrographs of biofilms formed by *P. aeruginosa* PU15 (scale bars 5 μm and 2 μm). Untreated biofilm, ZnO-NPs-treated biofilm, and meropenem–ZnO-NPs combination-treated biofilm.

**Figure 3 antibiotics-11-01374-f003:**
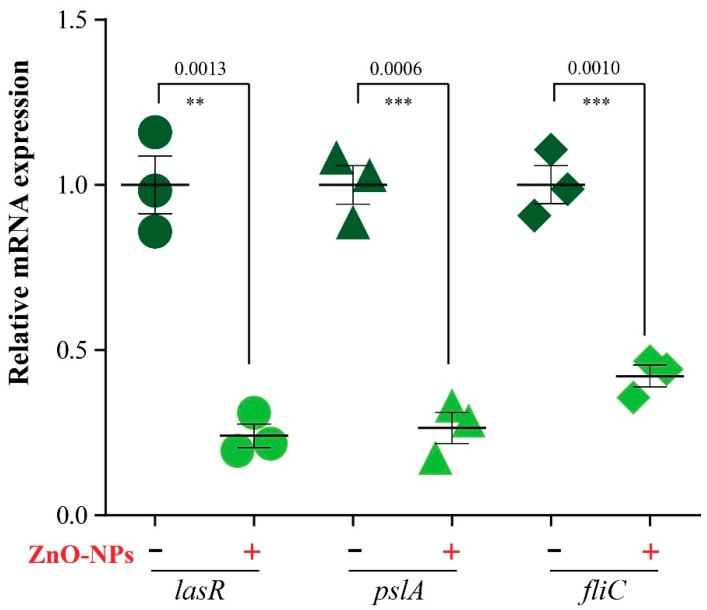
The effect of ZnO-NPs on the relative gene expression of biofilm genes (*lasR*, *fliC*, and *pslA*). The data represent the mRNA expression of each gene relative to *16S rRNA*. Results are expressed as Mean ± SEM from three independent experiments. ** *p* ≤ 0.01; *** *p* ≤ 0.001.

**Figure 4 antibiotics-11-01374-f004:**
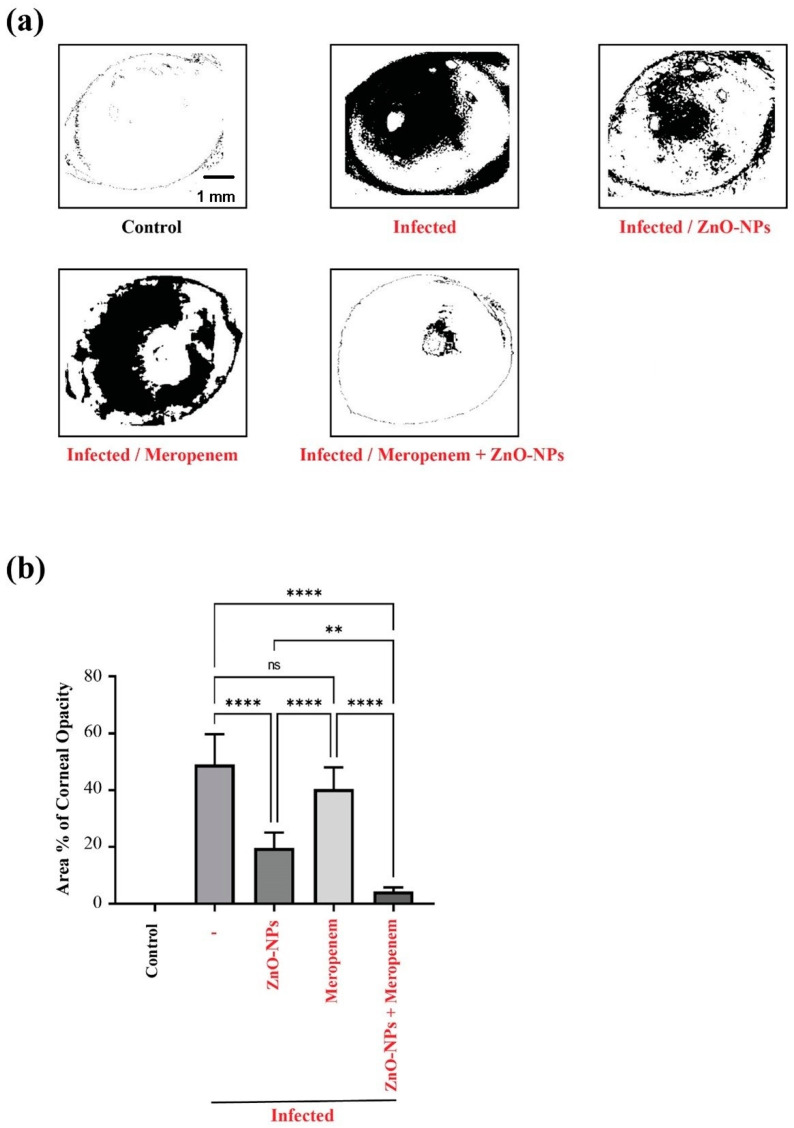
ZnO-NPs and meropenem mixture protects the rat cornea against pseudomonas-induced keratitis. (**a**) The Imagej-processed images of *P. aeruginosa*-infected eyes from different groups indicating focal lesions and corneal opacity (black areas), which represent a direct measure of the infection intensity. (**b**) Effect of meropenem and ZnO-NPs on the Area % of corneal opacity. Data are presented as Mean ± Standard Deviation (SD). ** *p* ≤ 0.01; **** *p* ≤ 0.0001.

**Table 1 antibiotics-11-01374-t001:** Antibiotic susceptibility of *P. aeruginosa* strains.

Tested Strains	Antibiotics and Diameter of Inhibition																			
	Imipenem (10 µg)		Meropenem (10 µg)		Levofloxacin (5 µg)		Ceftazidime (30 µg)		Gentamycin (10 µg)		Amikacin (30 µg)		Aztreonam (30 µg)		Piperacillin (100 µg)		Cefepime (30 µg)		Norfloxacin (10 µg)	
	R	S	R	S	R	S	R	S	R	S	R	S	R	S	R	S	R	S	R	S
	≤15	≥19	≤15	≥19	≤13	≥17	≤14	≥18	≤12	≥15	≤14	≥17	≤15	≥22	≤14	≥21	≤14	≥18	≤12	≥17
PS20	R		R		R		R		R		R		R		R		S		R	
PS21	R		R		R		R		R		R		S		R		R		R	
PS22	R		R		R		R		R		R		S		R		R		R	
PS23	R		R		R		R		S		R		R		R		R		R	
PS24	R		R		R		R		R		S		R		R		R		S	
PS25	R		R		R		R		R		R		R		R		R		S	
PS26	R		R		R		R		R		R		R		R		S		S	
PS27	R		R		R		S		R		R		R		R		S		R	
PS28	R		R		R		R		R		R		R		R		R		R	
PS29	R		R		R		R		R		R		R		R		R		R	
PS30	R		R		R		R		R		R		R		R		R		R	
PS32	R		R		R		R		R		R		S		R		R		S	
PS33	R		R		S		R		R		R		R		R		R		S	
Pp6	R		R		R		R		R		R		R		R		R		S	
Pp7	S		R		R		R		R		R		R		R		S		R	
Pp8	R		R		R		R		R		R		R		S		R		R	
Pp10	R		R		R		R		R		S		R		R		R		R	
Pp12	R		R		R		R		R		R		R		R		S		R	
Pp13	R		R		R		R		R		R		R		R		R		R	
PS35	R		R		R		R		R		R		R		R		R		R	
PS34	R		R		R		R		R		R		R		R		R		R	
PU17	R		R		R		R		R		R		S		R		R		R	
PC38	R		R		R		R		R		R		R		R		R		R	
PU15	S		R		R		R		R		R		R		R		R		R	
R %	91.6		100		96		96		96		91.6		83		96		79		75	

R: resistant; S: sensitive; R%: percentage of resistant strains.

**Table 2 antibiotics-11-01374-t002:** Antimicrobial synergism between meropenem and ZnO-NPs.

Antibacterial Agents	MIC (mg/mL)		FIC	FICI	Interpretation
	Alone	Combination			
ZnO-NPs	0.0640 ± 0.00462	0.0080 ± 0.00231	0.125	0.132	Synergy
Meropenem	0.5120 ± 0.01848	0.0040 ± 0.00058	0.007		

MIC values are expressed as Mean ± Standard Error of the Mean (SEM) from three independent experiments.

**Table 3 antibiotics-11-01374-t003:** Primers and RT-PCR thermal cycles.

Target Gene	Primer Sequences	Reverse Transcription	Primary Denaturation	Amplification (40 Cycles)			Reference
				Secondary Denaturation	Annealing (Optics On)	Extension	
* **16S-rRNA** *	GACGGGTGAGTAATGCCTA	50 °C 30 min	94 °C 15 min	94 °C 15 s	60 °C 30 s	72 °C 30 s	[57]
	CACTGGTGTTCCTTCCTATA						
* **lasR** *	AAGTGGAAAATTGGAGTGGAG						[58]
	GTAGTTGCCGACGACGATGAAG						
** *pslA* **	TCCCTACCTCAGCAGCAAGC						[32]
	TGTTGTAGCCGTAGCGTTTCTG						
* **fliC** *	TGAACGTGGCTACCAAGAACG						[40]
	TCTGCAGTTGCTTCACTTCGC						

## Data Availability

Not applicable.

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
