# Peer review of "Combination of Meropenem and Zinc Oxide Nanoparticles; Antimicrobial Synergism, Exaggerated Antibiofilm Activity, and Efficient Therapeutic Strategy against Bacterial Keratitis"

_antibiotics, 2022, doi:10.3390/antibiotics11101374_

Round 1

Reviewer 1 Report

This is an interesting study that extends what is known regarding antimicrobial actions of ZnO nanoparticles.

Several minor revisions are needed.

Figure 1 parts b and c should be replaced by a sample table. Part a is not necessary (or could be moved to the supplementary data). 

Line 128 The words "highest intense biofilm" need to be changed to a better description, such as thickest biofilm or greatest biofilm biomass.

Line 133. Figure 2 has been truncated. The legend does not need to refer to (a) as there is only one part to this figure.

Page 8. Panel a of Figure 5 needs at least one scale bar. The legend needs to explain that the black areas represent corneal opacity.

Line 286. The information given for the ZnO nanoparticles is insufficient. Include the Sigma catalogue number (e.g. 677450) and also the batch number of the nanoparticles used. The information given, i.e. MFCD00011300, is just the MDL number and is not useful.  (see https://www.sigmaaldrich.com/AU/en/product/aldrich/677450).

Line 381. State the total number of rats used.

Line 395. State the number of rats per group.

Line 406. How were data sets checked for normality before parametric statistical analyses were done?

Line 416. This has duplication and should read

"Our results provide support for the concept of improving carbapenem by combining it with ZnO-NPs."

The paper has a large number of language errors and needs considerable editing to address those. This includes using italics for names of bacteria. The discussion has suboptimal paragraph structure; all the one sentence paragraphs need to be merged.

Author Response

Dear editor,

We would like to thank the reviewers for their valuable comments and efforts to improve the quality of our manuscript. In the following, we highlighted the comments and concerns of the reviewers. We addressed responses specific for each comment bellow. Thank you very much.

Comments of Reviewer #1 and our responses for each comment

Comment

This is an interesting study that extends what is known regarding antimicrobial actions of ZnO nanoparticles.

Response

We thank the reviewer so much for finding our work interesting  

Comment

Figure 1 parts b and c should be replaced by a sample table. Part a is not necessary (or could be moved to the supplementary data). 

Response

We thank the reviewer for his comment, we agree with him, and we followed his recommendation and replaced the figure

Comment

Line 128 The words "highest intense biofilm" need to be changed to a better description, such as thickest biofilm or greatest biofilm biomass.

Response

Thanks for your comment, and we replaced it

Comment

Line 133. Figure 2 has been truncated. The legend does not need to refer to (a) as there is only one part to this figure.

Response

Thank you for your comment, and we fixed it

Comment

Page 8. Panel a of Figure 5 needs at least one scale bar. The legend needs to explain that the black areas represent corneal opacity.

Response

Thanks so much for careful notification, we inserted the required scale bar, and we completed the figure legend

Comment

Line 286. The information given for the ZnO nanoparticles is insufficient. Include the Sigma catalogue number (e.g. 677450) and also the batch number of the nanoparticles used. The information given, i.e. MFCD00011300, is just the MDL number and is not useful.  (see https://www.sigmaaldrich.com/AU/en/product/aldrich/677450).

Response

Thanks so much for your comment, and we corrected it

Comment

Line 381. State the total number of rats used.

Response

We included it

Comment

Line 395. State the number of rats per group.

Response

We included it

Comment

Line 406. How were data sets checked for normality before parametric statistical analyses were done?

Response

Data sets were checked for normality using Excel before parametric statistical analyses

Comment

Line 416. This has duplication and should read

"Our results provide support for the concept of improving carbapenem by combining it with ZnO-NPs."

Response

Thanks so much for your comment, and we removed the duplicated sentence

Comment

The paper has a large number of language errors and needs considerable editing to address those. This includes using italics for names of bacteria. The discussion has suboptimal paragraph structure; all the one sentence paragraphs need to be merged.

Response

We respect the reviewer comment, and we worked hard to improve the language of the manuscript

Reviewer 2 Report

Dear author

Thank you for your effort. Some notes, recommendations and suggestions:

Please modify writing in the method section, similarity is high.

Microorganisms nomenclature should be italic in all the text.

Line 75-81 references jumped from 19 to 25. Please revise references.

Line 96 Beginning of the phrase needs revision

Table S1 needs to be placed in text, it's essential not supplemental

Figure 1 notion needs revision. It looks like two results for the same tested material.

Figure 2 do you mean the columns in green are before treatment and the colorless column represents after treatment? Explain please

Line 192 lethal bacterial biofilm is a claim that definitely requires verification/reference 

Line 253 what temperature you kept your bacteria at?

Line 255: Did you immerse the antibiotic with disc, or you bought ready discs, please explain for the reader

Line 271 , line 293, I believe the concentration of the bacteria is 108 not 108. Line 312 the same. Revise

I believe liter notation is used in uppercase not lower case. please refer to author guidelines for submission to MDPI

Line 253 , symbol of celsius.

Line 270, check the space before 200

Line 276 remove repeated word

Line 277 , remove (exactly) please.

The resolution of SEM micrographs can be improved.

I have some questions and recommendation

Could you add MTT test for biofilm?

Could you exclude the synergism against Proteus species? Could not other strains ?

In the introduction part, could you please identify possible application of ZnO nanopartices? Is for wound/ complicated wound or burns? Could you run toxicity test on fibroblast,cells or retina since you tried animal studies as well.

Author Response

Dear editor,

We would like to thank the reviewers for their valuable comments and efforts to improve the quality of our manuscript. In the following, we highlighted the comments and concerns of the reviewers. We addressed responses specific for each comment bellow. Thank you very much.

Comments of Reviewer #2 and our responses for each comment

Comment

Thank you for your effort

Response

We thank the reviewer so much for his nice words  

Comment

Please modify writing in the method section, similarity is high.

Microorganisms nomenclature should be italic in all the text.

Response

We thank the reviewer for his comment, we agree with him, and we corrected in the Microorganisms nomenclature in the methods section

Comment

Line 75-81 references jumped from 19 to 25. Please revise references.

Response

Thanks for your comment, and we fixed this error

Comment

Line 96 Beginning of the phrase needs revision

Response

Thank you for your comment, and we fixed it

Comment

Table S1 needs to be placed in text, it's essential not supplemental

Response

we added it

Comment

Figure 1 notion needs revision. It looks like two results for the same tested material

Response

Thanks so much for your comment, and we replaced Figure 1

Comment

Figure 2 do you mean the columns in green are before treatment and the colorless column represents after treatment? Explain please

Response

Thanks for your comment, and we explained it in the figure caption

Comment

Line 192 lethal bacterial biofilm is a claim that definitely requires verification/reference 

Response

We refer in this statement to the serious nature of P. aeruginosa infection itself and we included reference 32

Comment

Line 253 what temperature you kept your bacteria at?

Response

Usually, we keep frozen stocks of our isolates at -80℃

Comment

Line 255: Did you immerse the antibiotic with disc, or you bought ready discs, please explain for the reader

Response

We purchased ready-made antibiotic discs from Oxoid Ltd., England

Comment

Line 271 , line 293, I believe the concentration of the bacteria is 108 not 108. Line 312 the same. Revise

Response

Thanks so much for careful notification, and we revised them

Comment

I believe liter notation is used in uppercase not lower case. please refer to author guidelines for submission to MDPI

Response

Thanks so much for careful notification, and we revised all the units according to the guidelines of the journal

Comment

Line 253 , symbol of celsius.

Line 270, check the space before 200

Line 276 remove repeated word

Line 277 , remove (exactly) please.

Response

Thanks so much for careful notification, all fixed.

Comment

The resolution of SEM micrographs can be improved.

Response

We increase the resolution

Comment

Could you add MTT test for biofilm?

Response

We performed the possible assays to quantify the biofilm intensity depending on the available kits and facilities in our lab but that is a good point, we wish we could do it

Comment

Could you exclude the synergism against Proteus species? Could not other strains?

Response

Very interesting point to start another project, I guess we have to consider other pathogens, the limitation will be the side effects of ZnO-NPs when administered systemically

Comment

In the introduction part, could you please identify possible application of ZnO nanopartices? Is for wound/ complicated wound or burns? Could you run toxicity test on fibroblast,cells or retina since you tried animal studies as well.

Response

We thank so much the reviewer for his ideas and recommendations, actually we built our approach on the effect of ZnO-NPs as a biofilm inhibitor and we chose a model for keratitis infection caused by P. aeruginosa, the problem due to limitation of facilities, we could perform some of the recommended experiments.

We shortlisted our introduction to focus on biofilm formed by P. aeruginosa and its role in developing resistance against Carbapenems